# H3K9me3 Levels Affect the Proliferation of Bovine Spermatogonial Stem Cells

**DOI:** 10.3390/ijms25179215

**Published:** 2024-08-25

**Authors:** Rui Yang, Boyang Zhang, Yueqi Wang, Yan Zhang, Yansen Zhao, Daozhen Jiang, Lanxin Chen, Bo Tang, Xueming Zhang

**Affiliations:** State Key Laboratory for Diagnosis and Treatment of Severe Zoonotic Infectious Diseases, Key Laboratory for Zoonosis Research of the Ministry of Education, College of Veterinary Medicine, Jilin University, Changchun 130062, China; ruiyang22@mails.jlu.edu.cn (R.Y.); zby23@mails.jle.edu.cn (B.Z.); yueqiw22@mails.jlu.edu.cn (Y.W.); z_yan22@mails.jlu.edu.cn (Y.Z.); yszhao21@mails.jlu.edu.cn (Y.Z.); jiangdz23@mails.jlu.edu.cn (D.J.); chenlx21@mails.jlu.edu.cn (L.C.)

**Keywords:** bovine, testes, spermatogonial stem cells, histone H3 lysine 9 trimethylation, cell proliferation

## Abstract

Spermatogonial stem cells (SSCs) possess the characteristics of self-renewal and differentiation, as well as the ability to generate functional sperm. Their unique stemness has broad applications in male infertility treatment and species preservation. In rodents, research on SSCs has been widely reported, but progress is slow in large livestock such as cattle and pigs due to long growth cycles, difficult proliferation in vitro, and significant species differences. Previously, we showed that histone 3 (H3) lysine 9 (K9) trimethylation (H3K9me3) is associated with the proliferation of bovine SSCs. Here, we isolated and purified SSCs from calf testicular tissues and investigated the impact of different H3K9me3 levels on the in vitro proliferation of bovine SSCs. The enriched SSCs eventually formed classical stem cell clones in vitro in our feeder-free culture system. These clones expressed glial cell-derived neurotrophic factor family receptor alpha-1 (GFRα1, specific marker for SSCs), NANOG (pluripotency protein), C-KIT (germ cell marker), and strong alkaline phosphatase (AKP) positivity. qRT-PCR analysis further showed that these clones expressed the pluripotency genes *NANOG* and *SOX2,* and the SSC-specific marker gene *GFRα1*. To investigate the dynamic relationship between H3K9me3 levels and SSC proliferation, H3K9me3 levels in bovine SSCs were first downregulated using the methyltransferase inhibitor, chaetocin, or transfection with the siRNA of H3K9 methyltransferase suppressor of variegation 3-9 homologue 1 (SUV39H1). The EDU (5-Ethynyl-2′-deoxyuridine) assay revealed that SSC proliferation was inhibited. Conversely, when H3K9me3 levels in bovine SSCs were upregulated by transfecting lysine demethylase 4D (KDM4D) siRNA, the EDU assay showed a promotion of cell proliferation. In summary, this study established a feeder-free culture system to obtain bovine SSCs and explored its effects on the proliferation of bovine SSCs by regulating H3K9me3 levels, laying the foundation for elucidating the regulatory mechanism underlying histone methylation modification in the proliferation of bovine SSCs.

## 1. Introduction

Spermatogonial stem cells (SSCs) are located on the basement membrane of the seminiferous tubules and possess the capability to self-renew and differentiate and can naturally transmit genetic information to offspring [1]. As the foundation of spermatogenesis, SSCs progress through various stages of spermatogenic cells by continual proliferation and differentiation and culminate in the production of mature sperm. Their unique stemness has broad potential applications in male infertility treatment and species preservation, particularly for high-yield, top-quality, and disease-resistant livestock breeding, as well as the germplasm preservation of endangered animals. The highly coordinated and complex process of male germline development is called spermatogenesis and is crucial for maintaining the reproductive ability of male mammals [2]. SSCs originate from prospermatogonia (ProSG, also named gonocytes) and develop into multiplying ProSG (M-ProSG), primary transitional ProSG (T1-ProSG), and secondary transitional ProSG (T2-ProSG). T2-ProSG can develop into SSCs and differentiated spermatogonia (Diff. SG) simultaneously [3]. The self-renewal and establishment of SSCs play a critical role in spermatogenesis, and their dysfunction can lead to SSC exhaustion, subsequently causing Sertoli cell-only syndrome (SCOS) [4,5]. The study of SSCs in rodents is currently more advanced [6,7], but there are still many challenges and unresolved issues regarding SSC research in large livestock like pigs and cattle, especially in terms of in vitro culture and transplantation technologies. The epigenetic regulation of SSCs is a key mechanism for ensuring normal male reproductive function. Epigenetics refers to non-genetic changes that affect gene expression without altering the DNA sequence itself, including DNA methylation, histone modifications, and non-coding RNA regulation. Among them, DNA methylation and histone modifications are the two major epigenetic events that not only impact the proliferation and differentiation of SSCs but also play important roles in their homeostasis.

Different from DNA methylation, histone modification regulates transcriptional activity in a more stable and dynamic manner at different stages. The DNA of germ cells is packaged into nucleosomes composed of histone 2A (H2A), histone 2B (H2B), histone 3 (H3), and histone 4 (H4), and undergoes covalent modifications such as methylation, acetylation, and ubiquitination [8]. Histone methylation occurs at the amino terminus of lysine and arginine residues, which differentially express histone lysine in the form of monomethylation (-me1), dimethylation (-me2), and trimethylation (-me3) [9]. H3K9me3 is one of the most important histone methylation modifications, showing a unique perinuclear distribution in mouse SSCs and a punctate distribution in differentiated spermatogonia, suggesting the presence of heterochromatin in the nuclear membrane. The characteristic perinuclear distribution of H3K9me3 is expected to serve as a new marker for SSCs [10]. The dynamic balance of histone methylation depends on the coordination of histone methyltransferases and demethylases. The SUV family of H3K9 methylases contains highly conserved SET domains, including suppressor of variegation 3-9 homologue 1/2 (SUV39H1/H2), euchromatic histone lysine methyltransferase 2 (EHMT2), euchromatic histone lysine methyltransferase 1 (EHMT1), and SET domain bifurcated histone lysine methyltransferase 1 (SETDB1). Loss of mouse *SUV39H* gene function delayed entry into the meiotic prophase and triggered significant apoptosis of spermatocytes in the mid-to-late pachytene [11]. The absence of EHMT2 caused the disappearance of H3K9me2 at E9.5 days and led to early embryonic death in mice [12]. The highly conserved lysine demethylase 4 (KDM4) subfamily (KDM4A-E, such as KDM4A, KDM4D, etc.) can demethylate di-methyl and tri-methyl residues of histone H3K9 and H3K36 [13]. Overexpression of *KDM4A* and other reprogramming factors could induce bovine mesenchymal stem cells to generate iPSCs, and their self-renewal ability could be extended to the 70th generation [14]. Microinjection of *KDM4D* mRNA into cloned buffalo and porcine embryos resulted in reduced H3K9me3 levels and increased pluripotency gene expression in the embryos and promoted the development of cloned embryos [15,16]. H3K9me3 also plays a critical regulatory role in the development of SSCs. Hexavalent chromium Cr (VI) can induce an increase in SETDB1 expression, upregulate H3K9me3 levels, and lead to SSC apoptosis, while melatonin can reverse the apoptosis of SSCs and decrease H3K9me3 levels, simultaneously reducing SETDB1 expression [17]. This suggests that SETDB1 may regulate H3K9me3 levels through histone modification. In SETDB1-deficient pig SSCs, both H3K9me3 levels and cellular proliferation were significantly reduced [18]. Similarly, it was found that knocking down SETDB1 impaired SSC proliferation and led to elevated intracellular ROS levels [19]. Additionally, the PTEN/AKT/FOXO1 pathway also participated in the regulation of SSC apoptosis by affecting the expression of SETDB1 [20].

Although the above studies provide credible evidence of the regulatory mechanisms of H3K9me3 on SSC proliferation, many potential molecular mechanisms remain unclear. In large livestock species, especially, it is unclear whether this conditioning mechanism still exists during bovine SSC development. Previously, we reported the molecular marker of bovine spermatogonia [21] and found that H3K9me3 was involved in the proliferation of bovine SSCs [22]. To further investigate the dynamic relationship between H3K9me3 levels and the proliferation of bovine SSCs, in this study, we first isolated and conducted in vitro culturing of bovine SSCs in a feeder-free system and then detected their proliferation after down/upregulating H3K9me3 levels using methyltransferase inhibitor chaetocin or transfection with the siRNA of *SUV39H1* or *KDM4D*. 

## 2. Results

### 2.1. Isolation, Purification, and Identification of Bovine SSCs

The unattached suspension cells were collected in the culture dish, and the somatic cells and part of the suspension cells began to adhere to the wall in vitro on Day 4 of the culture. On Day 9, small, scattered cell clones consisting of single cells appeared. On Day 15, the diameter and number of the cell clones gradually increased. On Day 30, the cell clones continuously enlarged and turned from transparent and bright to brown. The boundary lines of these clones were blurry and radial, and they were tightly attached to the culture dish (Figure 1a). The results of immunofluorescence staining demonstrated that the cell clones expressed GFRα1 (glial cell-derived neurotrophic factor family receptor alpha-1, a specific molecular marker for SSCs), NANOG (a pluripotency protein), and C-KIT (a major protein in proliferating germ cells) (Figure 1b), suggesting that the cell clones possess the characteristics of germ stem cells and exhibit pluripotency.

Since undifferentiated stem cells usually exhibit high alkaline phosphatase (AKP) activity, the cell clones on Day 15 and Day 30 of cultivation were stained with AKP, and the results of nuclear fast red staining showed that the cell colonies were strongly positive, appearing dark brown, while the surrounding cells showed light or non-colored staining (Figure 1c). qRT-PCR analysis revealed that the relative expression levels of the SSC marker gene *GFRα1*, as well as the pluripotency genes *NANOG* and *SOX2*, were significantly higher in the cell clones compared to the control group of bovine embryonic fibroblasts (BEFs), with *GAPDH* as the internal reference (Figure 1d). Together, the above results suggest that these cell clones are bovine SSCs.

### 2.2. SUV39H1 siRNA-Mediated Downregulation of H3K9me3 Levels Inhibit SSC Proliferation

The SSCs cultured for 20 d were transfected with *SUV39H1* siRNA. The qRT-PCR results showed a significant decrease in the relative expression level of *SUV39H1* mRNA in *si-SUV39H1*-transfected bovine SSCs compared to the control group and nonsense interference negative control (NC) group (Figure 2a). Consistently, western blot analysis demonstrated a significant downregulation of H3K9me3 levels in SSCs after transfection with *SUV39H1* siRNA compared to the control group and NC group (Figure 2b). After transfection, the proliferation of SSCs was detected using 5-Ethynyl-2′-deoxyuridine (EDU) staining. The results showed a significant reduction in cell proliferation in the *si-SUV39H1* group compared to the control and NC group (Figure 2c,d). Thus, the above data indicate that *SUV39H1* siRNA-mediated downregulation of H3K9me3 levels indeed inhibits the proliferation of bovine SSCs.

### 2.3. SUV39H1/2 Inhibitor Chaetocin Suppresses SSC Proliferation by Downregulating H3K9me3 Levels

To further confirm the above data, an SUV39H inhibitor was used for the culture and proliferation evaluation of bovine SSCs. Chaetocin is a specific inhibitor of histone H3K9 methyltransferase SUV39H1 and SUV39H2. Gradient concentrations of chaetocin were set at 0.01–0.05 µM, 0.08 µM, 0.1 µM, and 0.2 µM, and the experimental concentrations were screened using the CCK8 (Cell Counting Kit-8, Mei5bio, Beijing, China) assay. The cell viability analysis showed that none of the 0.01–0.05 µM chaetocin-treated SSCs showed significant differences compared with the control, while the high concentrations of chaetocin (0.08 µM, 0.1 µM, and 0.2 µM) were cytotoxic. Thus, three low concentrations (0.01 µM, 0.02 µM, and 0.03 µM) were selected for the subsequent experiments (Figure 3a). SSCs were treated with chaetocin at the three concentrations for 48 h, and qRT-PCR results showed that, with the untreated group as the control, chaetocin significantly reduced the mRNA expression levels of *SUV39H1* and *SUV39H2* (Figure 3b,c). Western blot assay further demonstrated that the chaetocin treatments resulted in a dramatic downregulation of H3K9me3 levels in bovine SSCs compared to the untreated control group (Figure 3d). Furthermore, the EDU staining showed that chaetocin inhibited the proliferation of SSCs at all three concentrations (Figure 3e). The percentages of EDU-positive cells in the chaetocin-treated groups were significantly lower than those in the control group (Figure 3f). Thus, the above results indicate that the SUV39H inhibitor chaetocin indeed downregulates H3K9me3 levels in bovine SSCs and thereby inhibits the proliferation of SSCs.

### 2.4. KDM4D siRNA-Mediated Upregulation of H3K9me3 Levels Promotes SSC Proliferation

After transfecting *KDM4D* siRNA into bovine SSCs (cultured for 20 days as before), qRT-PCR results showed a significant decrease in the relative expression level of *KDM4D* in bovine SSCs compared to the control and NC groups (Figure 4a). Western blot analysis demonstrated that the H3K9me3 levels in bovine SSCs were significantly upregulated after transfection with *KDM4D* siRNA (Figure 4b). EDU staining and statistical analysis showed that cell proliferation in the si-KDM4D-transfected group was significantly promoted compared with the control and NC groups (Figure 4c,d). These results indicate that *KDM4D* siRNA upregulates H3K9me3 levels in bovine SSCs and thereby promotes the proliferation of bovine SSCs.

## 3. Discussion

As the foundation of spermatogenesis, the proliferation and differentiation of SSCs are crucial for male reproductive capability. The in vitro culture of SSCs is a complex process involving multiple molecules acting synergistically to ensure that SSCs can maintain their self-renewal capacity and preserve their biological function. Current research is focused on the optimization of the culture system because SSCs tend toward apoptosis and differentiation during long-term culture. It has been reported that porcine SSCs could be maintained for up to 30 days in a serum-free condition [23]. In a culture system supplemented with GDNF, leukemia inhibitory factor (LIF), basic fibroblast growth factor (bFGF), and epidermal growth factor (EGF), mouse germ cells proliferated more than tenfold over a short period [24]. The addition of insulin-like growth factor 1 (IGF1) to this system resulted in continuous proliferation of porcine SSCs for 28 days without loss of undifferentiated spermatogonial phenotype [25]. In the culture system with GDNF + EGF + bFGF, bovine SSCs formed clones on Day 7 [26]. Furthermore, colony-stimulating factor 1 (CSF1) enhanced the self-renewal capacity of mouse SSCs without affecting the proliferation of non-SSCs [27]. 

Feeder cells are potential contaminations during sample collection, although their application has been shown to be helpful for the proliferation and maintenance of SSCs [28,29,30]. Additionally, the feeder cells might also alter the metabolic flux of SSCs. In a serum-free and feeder-free culture system, mouse SSCs could proliferate on laminin-coated dishes for over five months [31]. Transplantation of SSCs cultured in a feeder-free system into the testes of infertile mice produced normal and fertile offspring [32]. Reports have also shown that inoculating SSCs onto a hydrogel scaffold made from a decellularized testicular matrix (DTM) significantly enhanced the viability of SSCs compared to matrix gel or laminin-coated dishes [33]. These reports provided new insights into the feeder-free culture of SSCs. In this study, we isolated and cultured SSCs from calf testicular tissues, added LIF, bFGF, EGF, and GDNF to a feeder-free system to promote the proliferation and self-renewal of SSCs, and characterized the cultured bovine SSC clones. GFRα1 has been confirmed to be a marker protein for bovine SSCs [34,35]. NANOG and SOX2 are common pluripotent factors. Immunostaining showed that the cell clones were positive for GFRα1, NANOG, and SOX2. They also expressed the proliferative germ cell marker protein C-KIT. This evidence indicates that the cell clones were bovine SSCs and possessed pluripotency. Nuclear fast red staining of these cell clones showed strong AKP activity, implying that they are undifferentiated stem cells. Consistently, the cultured cell clones were further characterized from the mRNA level of *NANOG*, *SOX2*, and *GFRα1*. Collectively, the isolation, cultivation, and characterization of bovine SSCs laid a solid foundation for our next step to examine the influence of H3K9me3 levels on their in vitro proliferation.

Methylation of lysine residues at the histone tail is an epigenetic mark that is closely associated with spermatogenesis. Multiple histone methylation has been identified in SSCs and undifferentiated spermatogonia [36,37], among which H3K4 methylation promoted transcriptional activation while methylation of H3K9, H3K27, and H4K20 contributed to transcriptional inhibition [38,39]. H3K9me3 is enriched in mouse SSCs and exhibits a specific and complete perinuclear distribution pattern that disappears in differentiated spermatogonia cells [10]. It is also known that H3K9me3 is associated with chromatin structure and function [40], indicating that SSCs may contain heterochromatin in the nuclear membrane, and H3K9me3 is associated with the differentiation of SSCs. The perinuclear distribution of H3K9me3 may serve as a novel marker for male germ stem cells. H3K9me3 is dynamically involved in the regulation of spermatogenesis through methyltransferases and demethyltransferases. SUV39H1/H2 are methyltransferases that play crucial roles in mouse germ cell development and the cell cycle [11]. SUV39H2 could maintain the status of trophoblast stem cells and inhibit their differentiation by regulating H3K9 methylation status [41]. The KDM family, which contains the Jumonji C domain, is one of the demethylase families. KDM4D regulated DNA replication by reducing H3K9me3 levels to promote the formation of pre-initiating complexes [42], while RNA molecules regulated H3K9 methylation levels by affecting the binding of KDM4D to chromatin [43]. In addition, chaetocin is a specific inhibitor of the histone methyltransferases SUV39H1 and G9a, which can effectively downregulate H3K9me3 levels through pharmacological inhibition of methyltransferases [44]. Several articles have reported that inhibiting the expression of H3K9me3-specific methylase SETDB1 can induce apoptosis in cultured SSCs in vitro [17,18,19,20], suggesting that H3K9me3 might be associated with the maintenance of SSCs. Previously, our report indicated that H3K9me3 may be involved in the proliferation of bovine SSCs [22], but this still needs to be verified.

Therefore, in order to investigate the relationship between H3K9me3 levels and the proliferation of bovine SSCs, this study used both a chemical inhibitor and siRNA transfection to regulate H3K9me3 levels and examined the proliferation of bovine SSCs using EDU cell proliferation staining. The results indicated that the inhibitor chaetocin significantly reduced the expression levels of the methyltransferases *SUV39H1* and *SUV39H2*, and downregulated H3K9me3 levels in bovine SSCs. Transfection of *SUV39H1* siRNA silenced the *SUV39H1* gene in bovine SSCs and similarly reduced the level of H3K9me3 protein. The EDU staining results detected a significant inhibition of SSC proliferation under the conditions both of chaetocin treatment and *SUV39H1* siRNA transfection. Conversely, upregulating H3K9me3 levels with demethyltransferase *KDM4D* siRNA resulted in enhanced SSC proliferation.

At present, research on stem cell histone methylation is still limited, especially for large livestock. It is not yet clear through which pathways histone methylation regulates the proliferation of SSCs; however, based on the changing trend of H3K9me3 during spermatogenesis, we speculated that high levels of H3K9me3 may be associated with the maintenance of SSC stemness, while low levels of H3K9me3 may contribute to the differentiation of SSCs, thereby promoting cell proliferation. Additionally, histone methylation and DNA methylation in SSCs are closely correlated, and further research is needed to investigate whether other pathways are involved in the regulation of bovine SSCs. In summary, the epigenetic regulation of SSC development is a complex network involving multiple molecular mechanisms and signaling pathways. Understanding these mechanisms provides an important theoretical basis for studying the network of spermatogenesis and gene expression regulation, and has broad application prospects in stem cell biology, species conservation, and livestock genetic breeding.

## 4. Materials and Methods

### 4.1. Animal Tissues

The animal tissues from 1-day-old healthy Simmental calves were provided by Jilin Changchun Haoyue Islamic Meat Co., Ltd. (Changchun, China). The study was permitted by the Jilin University Institutional Animal Care and Use Committee for the use of animals/tissues (SY201903002).

### 4.2. Method Flowchart

The experiments were designed and conducted sequentially as shown in Figure 5.

### 4.3. Isolation and Purification of SSCs

Testicular tissues of 1-day-old calves were taken and soaked in a 75% alcohol solution for 10 min. Improvement of the methodology for isolation and purification of SSCs was based on our previously published article [22]. Briefly, excess tissues around the testis, such as epididymis and tunica albuginea, were removed aseptically and the testicular parenchyma was cut into about 1 cm^3^ and washed twice with phosphate buffer saline (PBS) containing 10% penicillin–streptomycin. Small pieces of testicular tissue were placed in a centrifuge tube and minced to a paste, left to stand for 5 min at room temperature, and then the supernatant was removed, and this operation was repeated twice to remove Leydig cells and the interstitial tissue. Subsequently, digestive solution I (F12 solution containing 1 mg/mL collagenase IV and 5 µg/mL DNase I) was added to the tube. The testicular tissues were digested at 37 °C for 30 min, and the supernatant was removed by centrifugation. Digesting solution II (F12 solution containing 1 mg/mL collagenase IV, 1 mg/mL hyaluronidase, 5 µg/mL DNase I, and 0.25% trypsin) was added to the tube and digestion was conducted at 37 °C for 25 min. The digestion was terminated by adding the termination solution containing fetal bovine serum (FBS) when single cells were observed. Then, the cells were washed two times with PBS and resuspended in SSC medium. The SSC culture medium was composed of F12 medium containing 2.5% FBS, bFGF (20 ng/mL), EGF (20 ng/mL), LIF (10 ng/mL), GDNF (30 ng/mL), β-mercaptoethanol (10 mg/mL), 1% non-essential amino acids (NEAA), and 1% penicillin–streptomycin. The cells were filtered through a 40 µm cell strainer and the unattached cell suspension was transferred into a new dish after 1.5 h of differential plating. The cells were continuously cultured for 1 h. Then, the cell suspension was collected in a new dish and the cells were incubated at 37 °C with 5% CO_2_. Collagenase IV, DNAse I, hyaluronidase, 0.25% trypsin solution, NEAA, penicillin–streptomycin, F12 medium, and β-mercaptoethanol were all obtained from Gibco (Grand Island, NY, USA), and bFGF, EGF, LIF, and GDNF were all sourced from Peprotch (Cranbury, NJ, USA).

### 4.4. Fluorescent Immunocytochemistry Staining

Cell coverslips were placed in the wells of the culture dishes before cultivation. They were taken out carefully for immunofluorescence staining after cultivation. The cell samples on the coverslips were washed with PBS 3 × 5 min, fixed in 4% paraformaldehyde solution for 15 min, and permeabilized with 0.5% TritonX-100 for 30 min. Subsequently, the cells were blocked with 3% bovine serum albumin (BSA) (*w*/*v*) for 1 h. After discarding the blocking solution, the primary antibodies were added dropwise to the cell coverslips (Table 1). The samples were kept in the wet box at 4 °C overnight. The next day, the cells were incubated with Alexa Fluor 594-conjugated goat anti-rabbit/mouse IgG (1:500; Proteintech, Rosemont, IL, USA) for 1 h. To visualize the nuclei, the cells were counterstained with 4′,6-diamidino-2-phenylindole (DAPI, Beyotim Biotechnology, Shanghai, China) at room temperature for 5 min. Finally, the cell samples on coverslips were covered with anti-quenching tablets and observed under a Nikon 80i fluorescence microscope (Nikon, Tokyo, Japan). Note that except for the step on adding primary antibody, the samples should be washed with PBS 3 × 5 min before each operation [45,46].

### 4.5. Alkaline Phosphatase (AKP) Staining

Based on the instructions on the AKP Staining Kit (Solarbio, G1480, Beijing, China), the cell culture medium was discarded, and the cells were washed with PBS. Then, the cells were fixed with 4% paraformaldehyde at room temperature for 15 min and washed with distilled water. Next, the cells were incubated with prepared ALP incubation solution (AS-BI staining solution/FBB staining solution = 1:1) for 15 min in a wet box, protected from light. Finally, the cells were washed in distilled water and then stained with the nuclear fast red staining solution for 3–5 min, washed, and examined microscopically [45,46].

### 4.6. qRT-PCR

Total RNA was extracted from bovine SSCs according to the instructions of the Animal RNA Extraction Kit (Beyotime, Shanghai, China), and cDNAs were obtained using the RNA Reverse Transcription Kit (TransGen Biotech, Beijing, China). Primers (Table 2) were designed according to the gene sequences in the NCBI database and synthesized by Sangon Biotech (Shanghai, China). Quantitative real-time polymerase chain reaction (qRT-PCR) systems of 20 µL were prepared and the reactions were carried out according to the manufacturer’s instructions (TransGen Biotech, Beijing, China). The experiments were repeated three times independently, and relative gene expressions were calculated according to the formula 2^−ΔΔCT^ [47]. Data were reported as mean ± SEM.

### 4.7. siRNA Transfection

Based on the sequences of *SUV39H1* (NM: 001046264.2) and *KDM4D* (XM: 015474601.2) provided by NCBI, siRNAs were synthesized by Sangon Biotech (Shanghai) Co., Ltd. A 20 µM volume of stock solutions of siRNA were prepared and stored at −20 °C. The siRNA sequences are listed in Table 3. Twenty-four hours prior to transfection, a fresh antibiotic-free culture medium was added and transfection was carried out when the cell density reached 80%. For a 6-well plate, 250 µL Opti serum-free medium (Gibco, Grand Island, NY, USA) and 5 µL siRNA were slowly added into one tube; into another tube was slowly added 250 µL Opti serum-free medium + 5 µL Lip2000 transfection reagent (Invitrogen, Los Angeles, CA, USA). Then, the transfection reagents in these two tubes were gently mixed together (lipofectamine 2000/siRNA = 1:1) and left at room temperature for 15–20 min. Next, the prepared transfection mixture was slowly and uniformly added into the wells containing cells. Finally, the cells were incubated for 4–6 h at 37 °C with 5% CO_2_. After transfection, the mixture was replaced with fresh medium and the cells were continuously incubated for 48 h [48].

### 4.8. Western Blot

Cell samples were collected and lysis solution (Beyotime, Shanghai, China) was added for full lysis. The supernatant was collected for centrifugation at 10,000–14,000× *g* for 10 min to obtain the extracted protein. The protein concentration was determined using the BCA Protein Quantification Kit (ThermoFisher Scientific, Shanghai, China). After quantification, the protein was mixed with 5× Loading Buffer (Beyotime, Shanghai, China) at a ratio of 1:4 and boiled for 10 min for denaturation. Next, the conditions were set at 75 V for 30 min for the concentration gel and 120 V for 1 h for the separation gel. The polyvinylidene fluoride (PVDF) films were cut to an appropriate size, moistened with methanol for 30 s, and then lightly placed on the gel. The transfer condition was set at 200 mA for 40 min. After blocking with 5% skimmed milk powder for 1.5 h, the PVDF films were placed in the primary antibody overnight at 4 °C. The analyzed protein was Histone 3 (H3) on lysine 9 (K9) trimethylated (H3K9me3) (1:1000, Cell Signaling Technology, Danvers, MA, USA). β-Actin (1:20,000, PROTEINTECH, Chicago, IL, USA) was used as an internal reference protein. The next day, the PVDF films were washed with TBST on a shaker for 3 × 10 min. Subsequently, horseradish peroxidase-conjugated goat anti-rabbit/mouse IgG (H + L) secondary antibody (Beyotime, Shanghai, China) was added, and the films were incubated at room temperature for 1.5 h. Next, ECL chemiluminescent agents (Thermo Scientific, Bannockburn, IL, USA) A and B were mixed at a ratio of 1:1 and added evenly and dropwise on the protein side of the PVDF films. The films were then put into an imager (GUANGYI, Guangzhou, Oi160, China) for exposure. Image J V1.8.0.112 software was used for relative expression greyscale-based analysis [49].

### 4.9. EDU Cell Proliferation Staining

Experiments were performed according to the instructions on the EDU Cell Proliferation Detection Kit (Ribobio, Guangzhou, China). In 96-well plates, for example, cells in each well were incubated for 2 h with 100 µL of 50 µM EDU medium (EDU solution/cell culture medium = 1:1000) and washed with PBS 2 × 5 min. Paraformaldehyde (4%) solution was used to fix the cells for 30 min at room temperature, and the fixative was then discarded. The specimens were incubated with 2 mg/mL glycine solution for 5 min to neutralize the excess aldehyde group and then washed with PBS. Then, 0.5% TritonX-100 solution was used to permeabilize the cells for 10 min, and the cells were rinsed with PBS 3 × 10 min. Next, 100 µL of Apollo staining reaction solution (prepared according to the instructions) was added to each well, the cells were incubated at room temperature in the dark for 30 min, and the staining reaction solution was discarded. Subsequently, 100 µL of 0.5% TritonX-100 permeabilization solution was added to the cells, and the samples were washed for 3 × 10 min each, followed by rinsing in PBS for 5 min. Finally, the cells were observed, counted, and imaged under a Nikon 80i fluorescence microscope [50].

### 4.10. CCK8 Cell Proliferation Assay

Cell suspensions (100 µL each, about 2000 cells) were added to each well of 96-well plates. After the cells were completely attached, different concentrations (0.01, 0.02, 0.03, 0.04, 0.05, 0.08, 0.1, and 0.2 µM) of the inhibitor chaetocin (Selleck, Shanghai, China) were added to the wells. After culturing the cells for 48 h, the medium was replaced with CCK8 solution (Mei5bio, Beijing, China) + culture medium at a ratio of 1:10 (approximately 100 µL per well). The wells seeded with cells and CCK8 solution and chaetocin were set as group A (dosing). The wells seeded without cells but containing medium and CCK8 solution were set as group A (blank). The wells seeded with cells and CCK8 solution but not chaetocin were set as group A (0 dosing). Finally, after incubation for 1–4 h in the cell incubator, absorbance was measured at 450 nm using a microplate reader (ReadMax, Shanghai, China), and cell viability was calculated based on the formula for cell viability (%) = [A (dosing) − A (blank)]/[A (0 dosing) − A (blank)] [51].

### 4.11. Statistical Analysis

Statistical significance was compared among the groups. SPSS 22.0 and GraphPad Prism 8.3.0 software were used for data analysis and graphing. The *t*-test was used for analysis of the significance of the univariates. * *p* < 0.05, ** *p* < 0.01, and *** *p* < 0.001 indicate that there is a statistical difference or that the difference is extremely significant.

## 5. Conclusions

The bovine cell clones obtained from the feeder-free system express the SSC-specific molecular marker GFRα1, pluripotency protein NANOG, and germ cell marker C-KIT. They are AKP-positive and express high levels of *GFRα1*, *NANOG*, and *SOX2* mRNA, indicating that the cultured cell clones are bovine SSCs. 

Treatment of bovine SSCs with the methyltransferase inhibitor chaetocin or transfection with methyltransferase *SUV39H1* siRNA both resulted in downregulation of H3K9me3 levels in bovine SSCs and inhibited their proliferation, while transfection of bovine SSCs with demethylase KDM4D siRNA effectively upregulated H3K9me3 levels in bovine SSCs and promoted SSC proliferation.

## Figures and Tables

**Figure 1 ijms-25-09215-f001:**
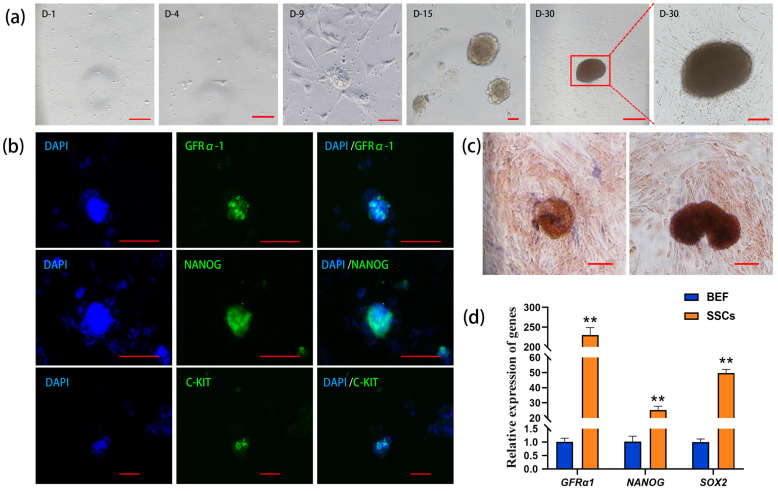
Isolation and identification of bovine spermatogonial stem cells (SSCs). (**a**) Formation of cell clones during in vitro culture (D: day); (**b**) immunofluorescence staining of GFRα1 (glial cell-derived neurotrophic factor family receptor alpha-1), NANOG (pluripotency marker), and C-KIT (marker of proliferating germ cells) in the cell clones (DAPI: 4′,6-diamidino-2-phenylindole counterstained nuclei; green: GFRα1, NANOG, and C-KIT positive staining; cycan: merged images); (**c**) alkaline phosphatase (AKP) staining of the cell clones; (**d**) qRT-PCR analysis of *GFRα1*, *NANOG*, and *SOX2* (pluripotency gene) in the cell clones with bovine embryonic fibroblasts (BEFs) as the control. Scale bars = 100 µm in (**a**–**c**), ** *p* < 0.01.

**Figure 2 ijms-25-09215-f002:**
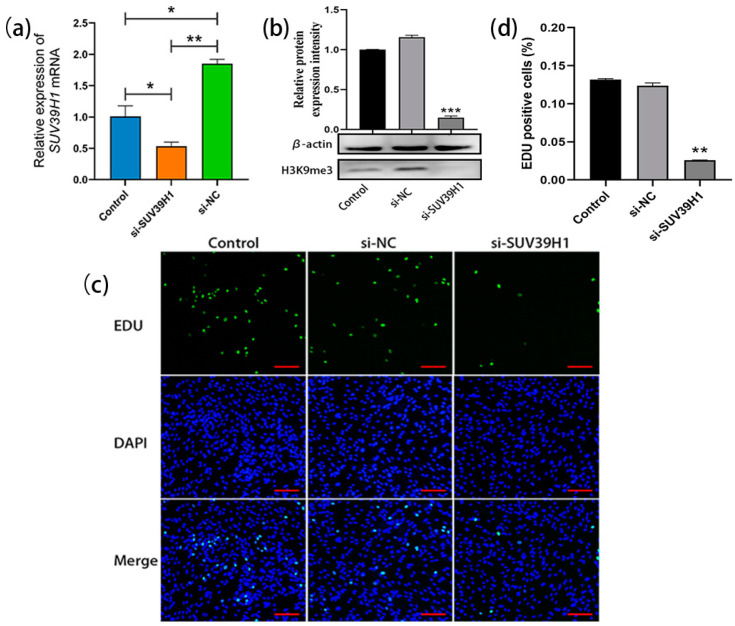
*SUV39H1* siRNA-mediated downregulation of H3K9me3 levels inhibited SSC proliferation. (**a**) The relative expression of *SUV39H1* mRNA in bovine SSCs (Control: non-transfected; si-SUV39H1: transfected with *SUV39H1* siRNA; si-NC: transfected with nonsense interference negative control); (**b**) H3K9me3 protein level detection using western blot; (**c**) EDU (5-Ethynyl-2′-deoxyuridine) cell proliferation staining (green: proliferating SSCs; DAPI: counterstained nuclei; merge: merged images of the EDU staining and DAPI-stained nuclei); (**d**) EDU-positive cell statistics. Scale bars = 100 µm; * *p* < 0.05, ** *p* < 0.01, and *** *p* < 0.001.

**Figure 3 ijms-25-09215-f003:**
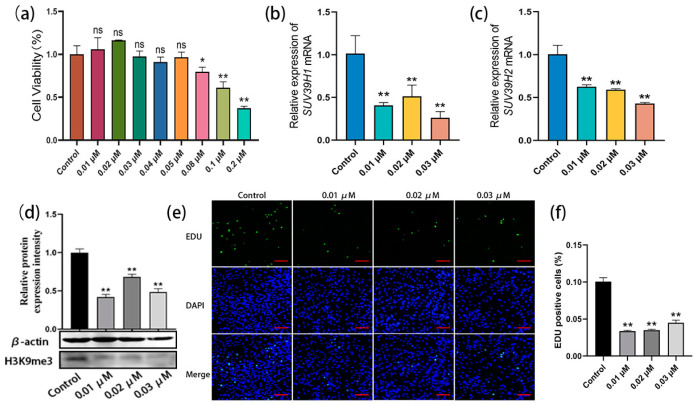
The SUV39H inhibitor chaetocin suppressed SSC proliferation by downregulating H3K9me3 levels. (**a**) Experimental concentrations of chaetocin were screened using Cell Counting Kit-8 (CCK8) assay; (**b**,**c**) relative mRNA expression of *SUV39H1* and *SUV39H2* in bovine SSCs; (**d**) H3K9me3 protein level detection using western blot; (**e**) EDU-staining of bovine SSCs treated with chaetocin at 0.01, 0.02, and 0.03 µM, respectively; (**f**) statistics on EDU-positive cell numbers. Scale bars = 100 µm; * *p* < 0.05 and ** *p* < 0.01.

**Figure 4 ijms-25-09215-f004:**
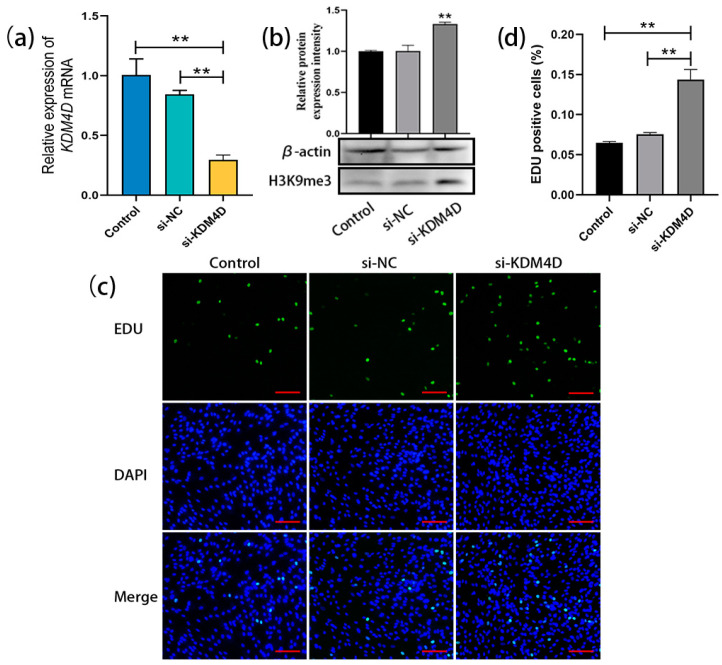
*KDM4D* siRNA-mediated upregulation of H3K9me3 levels promotes SSC proliferation. (**a**) The relative expression of *KDM4D* mRNA in bovine SSCs of the non-transfected control group, si-NC-transfected negative control, and si-KDM4D-transfected group; (**b**) H3K9me3 protein level detection using western blot; (**c**) EDU staining; (**d**) Statistics on EDU-positive cells. Scale bars = 100 µm; ** *p* < 0.01.

**Figure 5 ijms-25-09215-f005:**
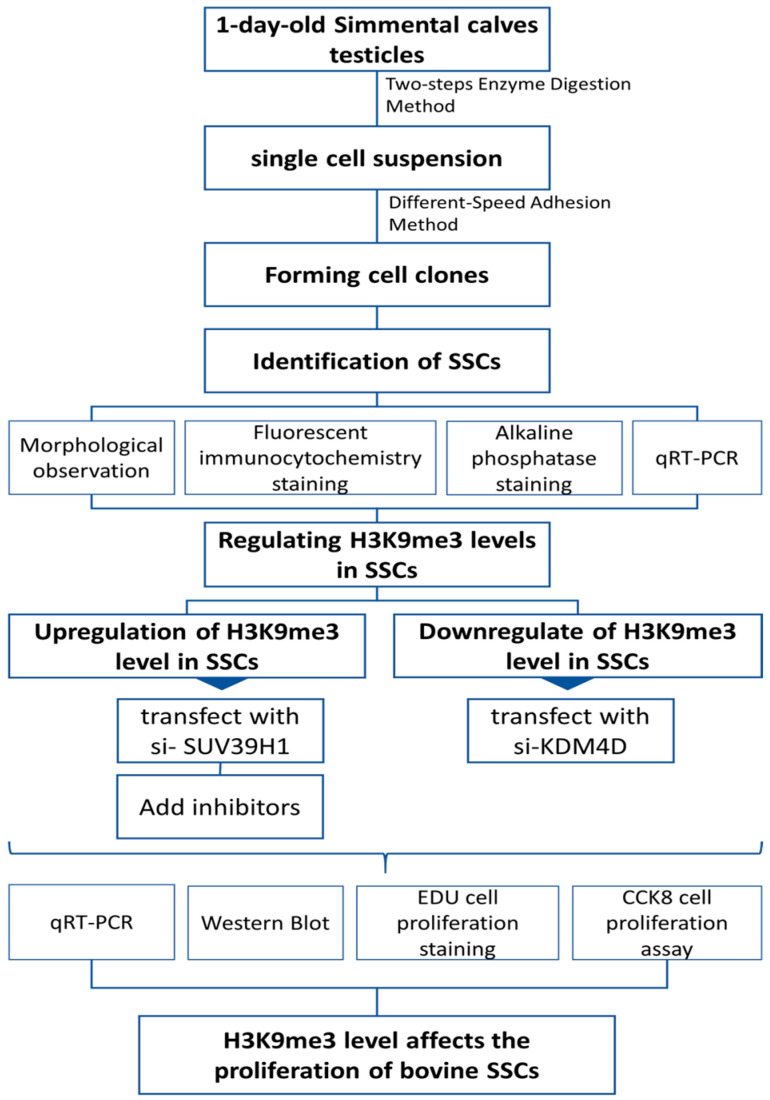
Experimental method flowchart.

**Table 1 ijms-25-09215-t001:** Information on the primary antibodies used for fluorescent immunocytochemistry staining.

Primary Antibody	Dilution Ratio	Species	Reagent Company
GFRα1	1:200	Rabbit	Abcam (Waltham, MA, USA)
NANOG	1:200	Rabbit	Abcam (Waltham, MA, USA)
C-KIT	1:100	Rabbit	Bioss (Beijing, China)

**Table 2 ijms-25-09215-t002:** Primers used for qRT-PCR.

Gene	Primer Sequence (5′→3′)	Fragment Size (bp)	Gene ID	Accession Number
*GAPDH*	F: CGGCACAGTCAAGGCAGAGAAC R: CCACATACTCAGCACCAGCATCAC	116	281181	NM_001034034.2
*GFRα1*	F: CTGACTGCCTCCTCGCCTAC R: AGGTCGTTTCCGCTGTTGTTG	118	534801	NM_001105411.1
*NANOG*	F: CACTGTCTCTCCTCTTCCCTCCTC R: TCTCTTCCTTCTCTGTGCTCTCCTC	122	538951	NM_001025344.1
*SOX2*	F: GGGCGGCAACCAGAAGAACAG R: CGCTTGCTGATCTCCGAGTTGTG	132	784383	NM_174580.3
*SUV39H1*	F: TGTGGACGCCGCCTATTACG R: GCCGCTCATCAAGGTTGTCTATG	104	523047	NM_001046264.2
*SUV39H2*	F: CACAGTGGATGCAGCTCGATATGG R: TGCTATTCGGGGAAGACGGGTATC	121	536936	NM_001037479.2
*KDM4D*	F: CGAGCAGGCTTGGCTAAGATAGTTC R: GGAGCGGAGCGGTTATTAAGATGTC	94	524768	XM_015474601.2

**Table 3 ijms-25-09215-t003:** Primers used for siRNA transfection.

Gene	Primer Sequence (5′–3′)
*SUV39H1*	F: CCAGACUCCGAGAGCACUUTT R: AAGUGCUCUCGGAGUCUGGTT
*SUV39H1-NC*	F: UUCUCCGAACGUGUCACGUTT R: ACGUGACACGUUCGGAGAATT
*KDM4D*	F: GCGAGUUCAUGGUGACCUUTT R: AAGGUCACCAUGAACUCGCTT
*KDM4D-NC*	F: UUCUCCGAACGUGUCACGUTT R: ACGUGACACGUUCGGAGAATT

## Data Availability

The data that support the findings of this study are available from the corresponding author, upon reasonable request.

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
