# Peer review of "H3K9me3 Levels Affect the Proliferation of Bovine Spermatogonial Stem Cells"

_ijms, 2024, doi:10.3390/ijms25179215_

Round 1

Reviewer 1 Report

Comments and Suggestions for Authors

The current study is a comprehensive investigation focused on elaboration of feeder-free stem cell culture engineering aimed at ectopic epigenetic rearrangement of bovine SSCs. The methodology and experimental protocol are evaluated by the Reviewer as complex, time- and labour-consuming and based on the proper assumptions related to statistical verification of the results obtained by Authors.

This research is the first to examine the effects of epigenetic modifier representing the class of histone methyltransferase inhibitors (termed Chaetocin) or siRNA-mediated cell transgenization (leading to silencing transcriptional activity of SUV39H1 gene or enhancing transcriptional activity of KDM4D gene) on the cytophysiological function of  SSCs, including their proliferative capability under ex vivo expansion conditions.

Conclusively, the current manuscript can be accepted for publication in its present form, which is very well written in English. 

Minor point of the Reviewer is as follows: The format of References has to be adjusted to the requirements of Int. J. Mol. Sci.

Reviewer 2 Report

Comments and Suggestions for Authors

In this study the attempts were made to culture bovine  spermatogonia stem cells  (SSCs) and analyze the regulatory effects of H3K9me3 on the proliferation of SSCs in vitro. The Reviewer suggests that the following comments would be helpful to improve the quality of the manuscript.

My comments are as follows:

1. The Reviewer suggests that the title should be re-phrased. Possible suggestion "Effects of H3K9me3 on the regulation of bovine spermatogonia stem cells  (SSCs)"

2. Should consider to include in vitro in the objectives (L101-104).

3. M&M

a) Give the appropriate references of the isolation and purification of SSCs,  AKP staining protocol, RT-qPCR analysis, etc.

b) Information is missing about the Western blotting analysis (L372-391) – appropriate reference, the analyzed proteins, the control protein, full names of abbreviation/synonyms, etc.

4. Results

a) More clarity is needed to interpret the data given in Figures 1-4. For examples, for  Figure 1, should consider to include that comparisons were performed between BEF (full name is required?) and SSCc (full name?) in the caption. For the other figures, should consider to indicate that comparisons were performed between the control and treatment. Provide the full names of all the abbreviation/synonyms in the captions of the figures. Give the number of samples analyzed for each treatment.

b) Figure 4. Reduced gene expression, in some cases, is concurrent with  lower protein, but in this study there was higher protein expression of KDM4D compared to its mRNA levels. Should consider to provide a possible explanation in the Discussion.

Comments on the Quality of English Language

The English needs to be improved, particularly in the Discussion.

Reviewer 3 Report

Comments and Suggestions for Authors

This study is very complex and provides major insights into the roles of H3K9me3 in the proliferation events in bovine Spermatogonial stem cells (SSCs).

My major concern lies principally in the complexity of the experimental approach which comprises two separate experiments. I must admit, I got lost in the complex methodology description which is why I highly recommend to design a scheme depicting the experimental design in detail. Also, I would suggest to clearly separate the methods used specifically for SSCs isolation and identification and for epigenomics.I was also wondering how did the authors confirm that following testicular isolation only pure SSCs were present in the culture. The presence of SSCs was undoubtedly verified by the fluorescent expression of target markers, but were any other cells present in the cultures? If yes, what types of cells?

I am curious about the rationale of the study. SSCs do have major implications in regenerative medicine, however, would it be feasible to perform an SSC transplant in cattle? If so, what would be the reasons?

All figures should be increased or added as supplementary material as it is very difficult to read particularly the graphs and Western blot images.

The authors shoul briefly discuss any limitations that may have affected the outcomes of the study. Also, what would be the implications of this study for further research in this area? Hypothetically, could H3K9me3 be considered a fertility marker in cattle?

On a minor note, please add the aims of the study into the Abstract section. Also, the references listed within the text should be written without superscript.
